# Genome Analysis of a Variant of *Streptomyces coelicolor* M145 with High Lipid Content and Poor Ability to Synthetize Antibiotics

**DOI:** 10.3390/microorganisms11061470

**Published:** 2023-05-31

**Authors:** Thierry Dulermo, Clara Lejeune, Ece Aybeke, Sonia Abreu, Jean Bleton, Michelle David, Ariane Deniset-Besseau, Pierre Chaminade, Annabelle Thibessard, Pierre Leblond, Marie-Joelle Virolle

**Affiliations:** 1Université Paris-Saclay, CNRS, CEA, Institute for Integrative Biology of the Cell (I2BC), Department of Microbiology, Group “Energetic Metabolism of *Streptomyces*”, 1 Avenue de la Terrasse, 91198 Gif-sur-Yvette, France; dulermothierry@gmail.com (T.D.); lejeune.clara@orange.fr (C.L.); michelle.david@i2bc.paris-saclay.fr (M.D.); 2Université Paris-Saclay, CNRS, CEA, Institut de Chimie Physique, UMR 8000, 91405 Orsay, France; eaybeke@gmail.com (E.A.); ariane.deniset@universite-paris-saclay.fr (A.D.-B.); 3Université Paris-Saclay, CNRS, CEA, Lip(Sys)2 (Lipides Systèmes Analytiques et Biologiques), UFR Pharmacie—Bâtiment Henri Moissan, 17 Avenue des Sciences, 91400 Orsay, France; sonia.abreu@universite-paris-saclay.fr (S.A.); jean.bleton@u-psud.fr (J.B.); pierre.chaminade@universite-paris-saclay.fr (P.C.); 4Université de Lorraine, INRAE, DynAMic, F-54000 Nancy, France; annabelle.thibessard@univ-lorraine.fr (A.T.); pierre.leblond@univ-lorraine.fr (P.L.)

**Keywords:** fatty acids/lipid biosynthesis/degradation, triacylglycerol, phosphatidylethanolamine, antibiotics, genome comparative analysis, genomic deletion, mobile genetic elements

## Abstract

*Streptomyces coelicolor* M145 is a model strain extensively studied to elucidate the regulation of antibiotic biosynthesis in *Streptomyces* species. This strain abundantly produces the blue polyketide antibiotic, actinorhodin (ACT), and has a low lipid content. In a process designed to delete the gene encoding the isocitrate lyase (*sco0982*) of the glyoxylate cycle, an unexpected variant of *S. coelicolor* was obtained besides *bona fide sco0982* deletion mutants. This variant produces 7- to 15-fold less ACT and has a 3-fold higher triacylglycerol and phosphatidylethanolamine content than the original strain. The genome of this variant was sequenced and revealed that 704 genes were deleted (9% of total number of genes) through deletions of various sizes accompanied by the massive loss of mobile genetic elements. Some deletions include genes whose absence could be related to the high total lipid content of this variant such as those encoding enzymes of the TCA and glyoxylate cycles, enzymes involved in nitrogen assimilation as well as enzymes belonging to some polyketide and possibly trehalose biosynthetic pathways. The characteristics of this deleted variant of *S. coelicolor* are consistent with the existence of the previously reported negative correlation existing between lipid content and antibiotic production in *Streptomyces* species.

## 1. Introduction

*Streptomyces coelicolor* M145 [1] abundantly produces some well-characterized specialized metabolites (cryptic PoyKetide/CPK, Calcium-Dependent Antibiotic/CDA, Undecylprodigiosin/RED and Actinorhodin/ACT) and is thus the model strain extensively studied by the scientific community to decipher the molecular basis of the regulation of antibiotic biosynthesis in *Streptomyces* species [2]. However, in contrast to most *Streptomyces* species, the production of these metabolites, and especially that of the blue polyketide Actinorhodin (ACT) [3], is rather constitutive in *S. coelicolor* because ACT biosynthesis is not repressed and is even stimulated in the condition of phosphate proficiency, at least upon growth on the commonly used solid medium R2YE [4]. Interestingly, a comparative proteomic analysis of *S. coelicolor* with *S. lividans*, a phylogenetically closely related strain, that possesses the same specialized metabolite biosynthetic pathways as *S. coelicolor* but does not express them, revealed that gluconeogenic enzymes were far more abundant in *S. coelicolor* than in *S. lividans,* whereas glycolytic enzymes showed an opposite trend [5]. This indicated that *S. lividans* has a glycolytic metabolism, generating acetylCoA in abundance, whereas *S. coelicolor*, that is characterized by an oxidative metabolism [4,6], is forced toward gluconeogenesis. Gluconeogenesis might depend on the metabolization of acetylCoA, originating from the degradation of fatty acids, via the glyoxylate cycle [7,8]. In *S. coelicolor*, enzymes of the glyoxylate cycle are *sco0982*/*ace*A encoding an isocitrate lyase cleaving isocitrate into glyoxylate and succinate and *sco0983*/*ace*B2 encoding a malate synthase that condenses glyoxylate and acetylCoA to yield malate. These genes are located upstream of *sco0984/aceB2* encoding a NAD+-dependent 3-hydroxyacyl-CoA dehydrogenase predicted to catalyze the third step of the β-oxydation of fatty acids. This enzyme oxidizes 3-hydroxyacyl-CoA into ketoacyl-CoA and generates NADH. The ketoacyl-CoA would then be cleaved by an acetyl-CoA-acyltransferase to yield acetyl-CoA. These three genes are located in divergence of *sco0981*, an orthologue of the transcriptional regulator AceR/IlcR of *E. coli* that represses the expression of genes encoding enzymes of the glyoxylate cycle in this species [9]. The full glyoxylate cycle is thus constituted by the isocitrate lyase, the malate synthase as well as by the malate dehydrogenase, the citrate synthase and the aconitase of the TCA cycle. The glyoxylate cycle is a short version of the TCA cycle that bypasses two of its decarboxylation reactions that yield NADH and thus support respiration and ATP generation [8]. Furthermore, with *sco0982*/*ace*A and *sco0983*/*ace*B2 being absent from *S. lividans* [10], we thought that the presence of these genes in *S. coelicolor* might contribute to the low total lipid content of this strain compared to *S. lividans* [11]. We thus decided to delete *sco0982* in *S. coelicolor* in order to test this hypothesis.

In the course of this deletion process, six dark-blue variants and one pale-blue variant (called TD) were obtained. The six dark blue variants had a level of ACT production similar to that of the original strain, whereas that of the TD variant was strongly reduced. Analysis in RT-PCR of the structure of the region of the target gene to be deleted was consistent with predictions for the six dark blue variants but that of the pale-blue TD variant was not. In consequence, its genomic DNA sequence was determined. This sequence revealed the occurrence of multiple deletions of various sizes (from 1 to 148 ORFs) including 9% of the total number of genes and among them the region encoding the enzymes of the glyoxylate cycle. This massive genome rearrangement was accompanied by the deletion of 60 of the 90 insertion sequences (ISs) present in the genome. Interestingly, the total lipid content of the TD variant was 2- to 3-fold higher, whereas its ACT production level was 7- to 15-fold lower than that of the original strain. Some of the possible causes of the unusual phenotypes of this variant in relation with its genomic rearrangements are proposed and discussed.

## 2. Materials and Methods

### 2.1. Strains and Culture Conditions

The strains used in this study are *S. coelicolor* M145 and a pale-blue variant of the latter (TD variant) whose ability to produce ACT and RED was strongly reduced and whose total lipid content was 2- to 3-fold higher than that of the original strain. These strains were grown on modified solid R2YE medium with no sucrose added and contained glucose (50 mM) as the main carbon source. The modified solid R2YE medium used was either limited (1 mM free Pi final originating from elements of the media, no K_2_HPO_4_ added) or proficient (5 mM free Pi final, 4 mM K_2_HPO_4_ added) in phosphate. The weight of lyophilized mycelial samples was determined to establish growth curves of the strains.

### 2.2. Constructions of KO Mutants of sco0982 in S. coelicolor

PCR-based REDIRECT technology was used to disrupt *sco0982* (*icl*) in *S. coelicolor* M145. *sco0982* was replaced in the genome by the *aac*(3)IV cassette [12] originating from the plasmid pHP45Ω*aac* (generous gift of J-L Pernodet). For this, fragments located in 5′ and 3′ of *sco0982* were amplified by PCR using chromosomal DNA from *S. coelicolor* M145 as a template and the primers listed in Appendix A, yielding an amplicon of 1508 bp and 1294 bp, respectively. PCR conditions used were the following: five cycles of 60 s at 68 °C, 60 s at 50 °C and 270 s at 72 °C; 25 cycles of 60 s at 96 °C, 60 s at 47 °C and 270 s at 72 °C in the presence of 5%DMSO. Amplification was carried out using Turbo Pfu polymerase from Stratagene. The PCR fragments and the *aac*(3)IV cassette conferring resistance to apramycin cut by *Sma*I (compatible with *Eco*RV ends) were ligated into pOSV400 cut by *Hind*III and *Bam*HI and carrying the hygromycin resistance gene [13], to yield pOSV400/*sco0982::aac(3)IV*. This plasmid was introduced into *S. coelicolor* M145 by conjugative transfer from a specific strain of *E. coli* [14]. After detection of the double recombinants (Apra^R^, Hygro^S^), the replacement of *sco0982* by the *aac(3)IV* cassette was confirmed by PCR in four Apra^R^/Hygro^S^ transformants using the oligos listed in Appendix A.

### 2.3. Construction of the Plasmid to Over-Express sco0984

The *sco0984* coding sequence was amplified by PCR using the genomic DNA of *S. lividans* TK24 and the oligos listed in Appendix A, in the conditions described above. The amplicon digested by *Hind*III and *Pst*I was cloned into pOSV554, a pSAM2-derived conjugative shuttle *E. coli*/*Streptomyces* vector, replicative in *E. coli* and integrative in *Streptomyces* and carrying the resistance to apramycine [15]. In this construct, the expression of *sco0984* is placed under the control of the *erm*E* promoter. The structure of the recombinant plasmids extracted from *E. coli* was checked by restriction analysis, and then the recombinant plasmid pOSV554/*sco0984* was conjugated from the *E. coli* strain S17-1 carrying an integrated RP4 derivative that carries the transfer genes of the broad host range IncP–type plasmid RP4 integrated in its chromosome [14,16].

### 2.4. Assay of Extracellular ACT Production

In order to assay extracellular ACT production of *S. coelicolor* M145 and its variant TD, 10^6^ spores of each strain were plated, in triplicate, on a cellophane disk laid down on the surface of 5 cm diameter plates of solid R2YE medium limited in Pi and incubated at 28 °C for 48 h, 72 h and 96 h (Figure 1A). The assay of extracellular ACT production was carried out as described in [17]. For this, the agar medium present under the cellophane disks of each replicate was collected, sheared and allowed to diffuse in 20 mL of water for 2 h at 4 °C. The first eluate was transferred into a new tube, and the same operation was repeated twice. The three eluates were pooled together and 3 mL of HCl (3 M) was added to 60 mL of the eluates and homogenized. The mixture was incubated on ice for 10 min to allow ACT precipitation. The precipitated ACT was collected by centrifugation (13,000× g for 10 min). Supernatants were discarded and the ACT pellets were re-suspended in 1 mL of KOH 1 M. Extracellular concentration of the ACT was measured as optical density at 640 nm in a Shimadzu UV-1800 spectrophotometer using KOH 1 M as blank. The concentration of the ACT in the extracts was calculated using the molar extinction coefficient of ACT at 640 nm = 25,320 L·mol^−1^·cm^−1^ and the concentration of the ACT was expressed in nanomoles of excreted ACT per mg of dry mycelium (Figure 1A).

### 2.5. Determination of Total Lipid Content Using Attenuated Total Reflectance–Fourier Transform Infrared Spectroscopy (ATR-FTIR) Measurements

In order to determine the lipid content of *S. coelicolor* M145 and its TD variant, ATR-FTIR was used as described previously [18]. For this, the mycelium of the two strains grown as described above was scrapped off the cellophane disks, lyophilized and hermetically stored in polypropylene tubes at room temperature in order to minimize the hydration of the samples. The lyophilized samples were then subjected to infrared (IR) spectroscopy using a Bruker Vertex 70 FT-IR spectrometer with a diamond ATR attachment (PIKE MIRacle crystal plate diamond ZnSe) and an MCT detector with a liquid nitrogen cooling system. Reference and sample analysis were carried out using 100 averaged scans conducted from 4000 cm^−1^ to 400 cm^−1^ with a spectral resolution of 4 cm^−1^. The recorded spectra showed several important absorption bands including an absorption band at 1740 cm^−1^ that is characteristic of the stretching of C=O esters function mainly present in esterified lipids and the absorption band at 1650 cm^−1^ that is characteristic of the stretching of C=O in the amide function, mainly present in proteins (amide I). To evaluate the lipid content of the strains, the ratio of the intensity of those 2 bands CO/amide 1 can be used [18]. The amount of esterified fatty acids present in the strains is thus proportional to this ratio.

### 2.6. Atomic Force Microscopic Measurements Coupled to IR Spectroscopy (AFM-IR)

In order to determine the localization of lipids in *S. coelicolor* M145 and its TD variant, AFM-IR was used as described previously in [19]. For this, 10^6^ spores of the two strains were spread on cellophane disks laid down on the surface of solid R2YE agar medium limited in Pi. A scalpel was used to make an incision on the cellophane disk to allow the insertion of CaF_2_ slides (transparent in the mid-infrared region) perpendicularly to the surface in order to allow the adhesion of mycelium onto the CaF_2_ slides and carry out the AFM-IR measurements with a nanoIR2s system (Anasys Instruments, Santa Barbara, CA, USA) bearing a tunable IR Quantum Cascade laser (MIRcat-QT, DAYLIGHT solutions, San Diego, CA, USA). The principle of AFM-IR microscope and the details of AFM-IR operation were reported by [20,21,22]. Briefly, the AFM-IR technique provides morphological, mechanical and chemical analysis including AFM imaging and IR spectroscopy with high resolution. The AFM-IR operation is based on probing the surface of a sample with an AFM tip and measuring the local thermal expansion of the sample resulting from local absorption of IR radiation. In this work, the IR beam was focused on the sample surface from the top. The experiments were carried out with the resonance-enhanced contact mode. In this mode, the repetition rate of the laser source matches with one of the contact mode resonances of the AFM cantilever. Performing resonance-enhanced AFM-IR with metal-coated AFM tips significantly increases the sensitivity of IR detection [21,23]. In this study, topography and IR maps were obtained simultaneously by scanning the sample surface with a gold-coated AFM tip (µmash, HQ:CSC38/CR-Au, k = 0.03 N/m and the cantilever resonance frequency f_res_ around 150 kHz) at room temperature. The IR absorption maps were acquired at 1740 cm^−1^. The topography and IR maps were collected with a 0.1 Hz scan rate in high resolution (500 × 500 pixel^2^) using contact mode. The AFM-IR images were treated using MountainLab 9 software from Digital Surf (Besançon, France), https://www.digitalsurf.com/fr/produits-solutions/multi-instrument/ accessed on 26 April 2023.

### 2.7. Esterified Fatty Acids Quantification Using GC-MS

In order to determine the nature of the fatty acids present in the lipid molecules of *S. coelicolor* M145 and its TD variant, a trans-esterification method adapted from Lepage and Roy [24] and described in [18] was used.

### 2.8. Comparative Analysis of the Genome of S. coelicolor M145 and of Its TD Variant

The genome of the pale-blue TD variant of *S. coelicolor* M145 was sequenced on the “Plateforme de Microbiologie Mutualisée” (P2M) of the Pasteur Institute (Paris). For this, an Illumina paired-end library was created from genomic DNA fragments using the Nextera XT kit. Paired-end reads were generated using a NextSeq500 sequencing (Illumina, San Diego, CA, USA) instrument. The coverage of the paired-end reads had an average of 100×. The sequencing reads were trimmed (low Phred score base trimming, exogenous oligonucleotide clipping, sequencing error correction, read coverage homogenization) and aligned on the reference *S. coelicolor* A3(2) reference genome (AL645882) retrieved from NCBI on the Geneious platform (Geneious Prime 2022.0.1, Biomatters Ltd., Auckland, New Zealand) using its mapper algorithm (parameters medium-low sensitivity, 2 iterations, minimum mapping quality of 20). Regions with a null coverage could result from either the true absence of the locus in the TD variant or to local properties leading to sequencing difficulties. To assess the deletion of the interesting loci, the low-covered regions were searched in the contigs formed de novo from the sequencing reads; those occurring within a contig were considered to reveal a true deletion. The sequencing reads of the genome of the TD variant (raw data) were submitted to SRA under the reference SUB12982887, on 23 March 2023.

## 3. Results

### 3.1. A Pale-Blue Variant of S. coelicolor Weakly Produces ACT and RED and Has a High Triacylglycerol and Phosphatidylethanolamine Content

A visual inspection of the color of the seven colonies obtained in the course of the deletion process of *sco0982* in *S. coelicolor* indicated that six of them were producing ACT at a similar level as the original strain and the chromosomal structure of these clones was consistent with predictions.

In contrast, the pale-blue variant of *S. coelicolor*, called TD, produces 7- to 15-fold less ACT than the original strain in the condition of Pi limitation (Figure 1A), whereas ACT production was undetectable in condition of Pi proficiency (Figure 1B). Furthermore, when plated at the same density of 10^6^ spores per plate, the TD variant reproductively yielded higher biomass than the original strain (Figure 1A and Figure 2).

Because we previously demonstrated the existence of a reverse correlation between total lipid content and antibiotic production [4,11,25], we used Fourier Transformed Infrared Spectroscopy (FTIR) to determine the CO/amide I ratio, that gave an indication of the total lipid content of the parental strain, of the TD variant and of the *bona fide* deletion mutants of *sco0982*.

After 48 h, the CO/amide I ratio of the TD variant was over twofold higher than that of the original strain (Figure 2) indicating that it has a higher total lipid content than the original strain. Interestingly, the CO/amide I ratio of the TD variant decreased over time (from 60 h till 96 h) indicating a reduction in its lipid content, whereas this ratio and thus the lipid content remained rather stable in the original strain (Figure 2).

In contrast, the six *bona fide* deletion mutants of *sco0982* were shown to have a CO/amide I ratio approximately 20% higher than that of the original strain after 72 h of cultivation, indicating only a slightly higher total lipid content (Appendix A). These data thus indicated that *sco0982* could contribute to the low total lipid content of *S. coelicolor* M145 but to a rather small extent.

In order to determine the local repartition of the lipids in the mycelial fragments of the parental strain and the TD variant, atomic force microscopy-based infrared spectroscopy analyses (AFM-IR) were carried out as described in the Section 2. For this, the IR laser wavenumber was fixed to 1740 cm^−1^ (λ specific of the C=O functions present in esterified lipids) and local IR absorption maps and the topography were simultaneously recorded. The topographic images of Figure 3a,b show that the mycelial fragments of the TD variant were less regular than that of the original strain and parts of these filaments were blistered. These filaments were also thinner than those of the original strain and this might indicate an osmotic issue. The corresponding IR maps (Figure 3a’,b’) revealed that the TD variant exhibited a multitude of red patches in its cytoplasm, some of them coinciding with the blisters, whereas no such patches were detected in the original strain. Those red patches indicated the presence of TriAcylGlycerol (TAG) inclusions as reported previously for other *Streptomyces* species [19].

In order to confirm the nature of the lipids accumulated in the TD variant, the analysis of the total lipid content of the two strains was achieved in LC/MS. Results shown in Figure 4 confirmed that the TD variant bears a 3.25-fold higher TAG content than the original strain as well as a 5-fold and 1.2-fold higher DiAcylGlycerol content, DAG1,3 and DAG1,2, respectively, than the original strain. This study also revealed that the TD variant also bears a 2.84-fold higher PhosphatidylEtanomlamine (PE) content than the original strain. This difference in the lipidic content likely contributes to the higher biomass weight of this variant compared to the original strain (Figure 1A and Figure 2). Furthermore, the higher content in free fatty acids (FA) and MonoAcyl Glycerol (MAG) of the original strain compared to the TD variant indicated that TAG and PE biosynthesis could not go to completion in the original strain. Interestingly, this analysis also revealed that the TD variant produces two- to threefold less undecylprodigiosine (RED) and oxidized RED than the original strain (Figure 4).

To determine the nature of the fatty acids (FA) present in the lipidic molecules of the two strains more precisely, fatty acid methyl esters (FAMEs) were prepared from the two strains and their structure was determined in GC/MS. Figure 5 indicated that the fatty acids C16:0 iso (4.43-fold), C15:0/C15:0 iso (2-fold), C17:0 iso (2-fold) and C17:0 anteiso (1,4-fold) were more abundant in the TD variant than in the original strain and the fatty acid C14:0 iso was detected in the TD variant but not in the original strain.

### 3.2. Genomic Structure of the Pale-Blue TD Variant of S. coelicolor

Because the structure of the region of the target gene to be deleted (*sco0982)* could not be clearly established by PCR in the TD variant, its genomic DNA sequence was determined and compared to that of the parental strain. This analysis revealed the presence of numerous deletions including one to several genes (1 to 148) as shown in Appendix A. In total, 704 genes were deleted (i.e., 9% of the total number of genes):-A total of 32 were deleted individually (Appendix A).-A total of 263 deleted genes were part of large deletions including the terminal arms of the chromosome (spanning over 101 genes: from SCO0001 to SCO0081 and SCO7827 to SCO7846). This was probably associated with circularization of the chromosome, a rearrangement event commonly observed in *Streptomyces* grown in laboratory conditions [26,27].-A total of 386 genes were deleted through 16 large deletions resulting from the loss of mobile genetic elements (MGEs) such as phages, plasmids or integrated conjugative elements (AICEs), etc.-A total of 60 of the 90 insertion sequences (ISs) (2/3) present in the strain were deleted and among them 21 ISs (including 2 ISs composed of 2 genes, therefore 23 genes) had probably been excised autonomously and 37 were carried by the large deletions mentioned above.

The TD variant is thus characterized by a massive loss of MGEs. This could result from a severe state of oxidative stress, because oxidative DNA damages are known to favor the excision of MGEs through the induction of the SOS stress response [28,29]. The loss of MGEs may give the TD variant a greater genomic stability than its parental strain.

#### 3.2.1. Deletion of the Region Encompassing Genes of the Glyoxylate Cycle

We first noticed that the region containing the target gene to be disrupted, *sco0982* (isocitrate lyase) was deleted in the TD variant. This deletion spans 22 genes from *sco0979* to *sco1000* and thus contains genes encoding proteins of the glyoxylate cycle: *sco0982* (isocitrate lyase), *sco0983* (malate synthase) and its regulator s*co0981* AceR/Ilc [8]. This deletion also includes *sco0980* coding for an acylCoA hydrolase involved in the regulation of fatty acids metabolism and *sco0984*, a 3-hydroxybutyryl-CoA dehydrogenase. The over-expression of *sco0984* in *S. lividans* was shown to lead to a strong reduction in the total lipid content of this strain (Appendix A) consistently with its predicted function. The deletion of all these genes is likely to slow down fatty acid/lipid degradation and thus result in lipid accumulation. This deleted region also includes the genes *sco0985* (5 methyltetrahydropteroyltriglutamate/homocysteineS-methyltransferase involved in methionine biosynthesis), *sco0988* (acetyltransferase), *sco0992* (cysteine synthase), *sco0995* (methyltransferase), *sco0999* (superoxide dismutase) and *sco0996-97-98* encoding an iron uptake system. The deletion of the *sco0996-97-98* operon might reduce iron supply and, interestingly, some reports in the literature mention that iron limitation does trigger lipid biosynthesis/accumulation [30,31].

#### 3.2.2. Deletion of Two Pairs of Genes Encoding Proteins of the TCA Cycle

Two pairs of genes *(sco4594/95* and *sco6269/70)* encoding α and β subunits of putative 2-oxoglutarate ferredoxin oxidoreductases, enzymes of the TCA cycle involved in the conversion of alpha ketoglutarate into succinylCoA, were deleted in the TD variant. The deletion of these genes most likely impairs or strongly reduces the activity of the TCA cycle and thus the assimilation of acetylCoA that could then be stored as fatty acids and lipids.

#### 3.2.3. Deletion of Genes Encoding Proteins Involved in Nitrogen Assimilation

Genes encoding the subunits α/NarG3, β/NarH3, δ/NarJ3 and γ/NarI3 (*sco4947-48-49-50*) of the nitrate reductase [32] involved in nitrogen assimilation were also deleted in the TD variant. The deletion of these genes is likely to aggravate the poor ability of *S. coelicolor* to assimilate nitrogen [33], resulting in nitrogen stress even if the *sco6532-6535* operon encoding subunits of another nitrate reductase (Nar I, J, H and G) remained intact in the TD variant. Because nitrogen stress is known to be an important trigger of carbon storage in the form of lipids and especially of TAG in various microorganisms [34,35,36], the deletion of these genes is likely to constitute a major contribution to the high total lipid content of the TD variant.

#### 3.2.4. Deletion of Genes Involved in the Biosynthesis of Specialized Metabolites

The largest deleted region *sco6806-sco6953 (148 genes)* encompasses the arseno polyketide biosynthetic pathway *(sco6811-sco6837)* [37]. This biosynthetic pathway is, together with that of ACT, one of the most highly expressed biosynthetic pathways of specialized metabolites in *S. coelicolor* [5]. This arseno polyketide was thus proposed, as ACT, to be involved in the resistance to oxidative stress. However, in contrast to ACT, this function was not demonstrated for the arseno polyketide and we ignore whether the deletion of this pathway would lead to an enhanced total lipid content. Furthermore, the genes *sco6273-74* encoding the type I polyketide synthase of the coelimycin biosynthetic pathway were also deleted in the TD variant. Interestingly, the deletion of this cluster was indeed shown to lead to an increase of the total lipid content of *S. coelicolor* M145 (manuscript in preparation).

#### 3.2.5. Deletion of sco7335 Encoding a Putative Amylase

*Sco7335* is located at the beginning of an operon involved in trehalose biosynthesis [38]. The deletion of this gene might impair the expression of this operon and/or trehalose biosynthesis. If that is the case, this would spare carbon/glucose for lipid biosynthesis as shown in *Yarrovia lipolytica* [39].

#### 3.2.6. Deletion of Genes Encoding Proteins Involved in Lipid Biosynthesis

Two of the nine genes of *S. coelicolor* M145 encoding putative acyl/acetyl/propionyl CoA carboxylases, *sco4921* and *sco6271*, were deleted in the TD variant. These enzymes, that catalyze the first step of fatty acid biosynthesis, are usually constituted by two subunits: α and β. In *S. coelicolor*, they include SCO2445/44, SCO2776/77, SCO4380/81 as well as SCO4921 (AccA2, acyl-CoA carboxylase subunit α), SCO4926 (PccB, propionyl-CoA carboxylase subunit β) and SCO6271 (Acc1, acyl-CoA carboxylase subunit β). AcetylCoA carboxylase catalyzes the carboxylation of acetyl-CoA to produce malonyl-CoA yielding straight/iso fatty acids. Propionyl CoA carboxylase catalyzes the carboxylation of propionyl-CoA to methylmalonyl-CoA yielding branched/ante-iso fatty acids (Figure 5).

SCO4921 and SCO6271 might constitute the α and β subunits of a carboxylase complex that would be absent in the TD variant. This leaves six genes encoding three functional carboxylase complexes, SCO2445/44, SCO2776/77 and SCO4380/81. The deletion of genes encoding enzymes catalyzing the first step of fatty acid biosynthesis (acyl/acetyl/propionyl CoA carboxylases) in this variant that bears a high TAG and PE content was unexpected but it might be necessary to limit fatty acids/lipids biosynthesis in a genetic context where the biosynthesis of these molecules is greatly up-regulated as a consequence of some of the deletions mentioned above.

Furthermore, we noticed the deletion, in the TD variant, of *sco6832*, one of the two genes encoding an L-methylmalonyl-CoA mutase. This enzyme catalyzes the conversion of L-methylmalonyl-CoA, originating from propionyl-CoA, into succinyl-CoA. We cannot exclude that the deletion of *sco6832* might lead to an enhanced availability of propionyl-CoA (C3) used as starter for the biosynthesis of lipids constituted by fatty acids with an uneven number of carbons (C15 and C17) that were more abundant in the TD variant than in the original strain as shown in Figure 5.

Many more genes encoding proteins with unknown or even known functions or transporters or transcriptional regulators, etc., were deleted in the TD variant, and we cannot exclude that some of them also contribute to the high lipid content of the latter.

## 4. Discussion

In this work we report the characteristics of the TD variant of *S. coelicolor* that has a high total lipid content and a poor ability to synthetize the ACT and RED antibiotics, and attempts were made to relate these unusual characteristics to its genomic content.

Among the possible causes of the enhanced ability of the TD variant to store acetylCoA as TAG and PE, we propose (1) the deletion of genes encoding enzymes of the TCA cycle that would either impair or reduce its functioning leading to an accumulation of acetyl-CoA stored as lipids; (2) the deletion of genes of the glyoxylate cycle and of *sco0984* that would reduce fatty acids/lipids degradation [7,8]; (3) the severe nitrogen stress resulting from the deletion of genes encoding the subunits of the nitrate reductase (SCO4947-48-49-50) that is known to be a powerful trigger of carbon storage in the form of lipids in various microorganisms [34,36,40]; (4) the deletion of biosynthetic pathways directing the biosynthesis of specialized metabolites of the polyketide family that potentially constitute competing acetyl-CoA drain; and (5) the possibly impaired expression of the trehalose operon or impaired trehalose biosynthesis due to the deletion of one of the first genes of this operon. Trehalose biosynthesis derives a fraction of the glucose pool and thus limits its use to generate acetyl-CoA for lipid biosynthesis.

However, a most important question is what could have been the initial triggering event responsible for the high genetic instability of the TD variant. Because the excision of MGE is known to be provoked by oxidative stress [28,29], it might be the deletion of the arseno polyketide pathway that would be responsible for the severe raise in oxidative stress leading to high genetic instability. This genetic instability would have selected events promoting the storage of acetyl-CoA as lipids to reduce the activation of the oxidative metabolism and thus the generation of oxidative stress [4,41,42]. Oxidative stress is thus most probably at a lower level in the TD variant than in the original strain. Because ACT was previously shown to play a role in the resistance to oxidative stress [5,43], oxidative stress is likely to be an important trigger of ACT biosynthesis (manuscript in preparation). Consequently, a weaker oxidative stress could explain the lower ACT production of the TD variant compared to the original strain. Nonetheless, the TD variant still produces ACT at a low level in the condition of Pi limitation (Figure 1A), a condition known to promote the activation of the oxidative metabolism and thus the generation of oxidative stress [44,45], whereas ACT production is undetectable in the condition of Pi proficiency (Figure 1B). The remaining low level of ACT production in the TD variant could be due to a low level of oxidative stress due the very active fatty acids/lipids biosynthesis taking place in the TD variant. Indeed, fatty acid biosynthesis involves Fatty Acid Synthases (FASs), enzymes that require NADPH as a co-factor and generate NADP+. This biosynthetic process, being highly active in the TD variant, might consume high amounts of NADPH. With NADPH also being a co-factor of thioredoxin reductases, and enzymes playing a crucial role in the resistance to oxidative stress, a very active NADPH-dependent fatty acid biosynthesis might thus lead to a reduced NADPH availability resulting in oxidative stress. Furthermore, the deletion of the superoxide dismutase encoding gene, *sco0999,* as well as of genes encoding enzymes of the glyoxylate cycle, might also contribute to the weak oxidative stress of the TD variant. Indeed some reports in the literature mention a role of the glyoxylate cycle in the resistance to oxidative stress [46,47].

Alternatively or jointly, because numerous reports in the literature mentioned that DNA hypomethylation is correlated with genomic instability [48,49,50], we cannot exclude that the deletion of genes encoding DNA methylases (*sco5331, sco6844* and *sco6885*) played a role in the genetic instability that led to the generation of the TD variant.

In conclusion, it should be stressed that the study of the TD variant of *S. coelicolor* confirms the existence of the previously reported negative correlation existing between total lipid content and antibiotic production in *Streptomyces* species [4,11,25].

## Figures and Tables

**Figure 1 microorganisms-11-01470-f001:**
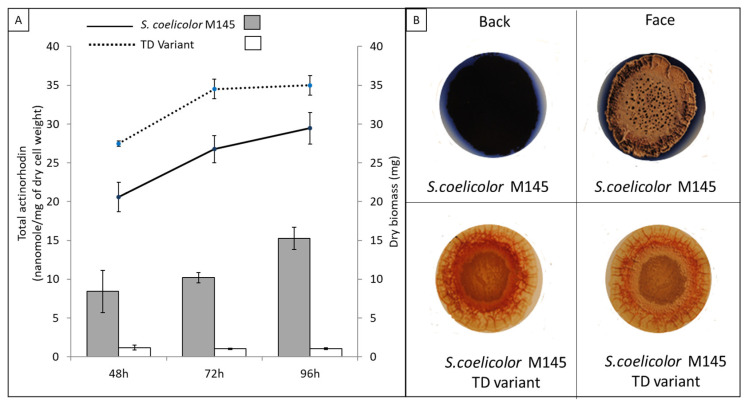
Characteristics of *S. coelicolor* M145 and its TD variant. (**A**) Quantification of extracellular ACT produced by *S. coelicolor* M145 (grey histograms) and its TD variant (white histograms) and assessment of dry biomass of *S. coelicolor* M145 (black continuous line) and its TD variant (black dotted line, grown for 48 h, 72 h and 96 h at 28 °C on solid R2YE medium limited in Pi (1 mM Pi)). (**B**) Pictures of plates of *S. coelicolor* M145 and its TD variant grown for 72 h at 28 °C on solid R2YE medium proficient in Pi (5 mM Pi). The *S. coelicolor* M145 plates are deep blue indicating strong ACT production, whereas the plates of the TD variant are colorless indicating the absence of ACT production.

**Figure 2 microorganisms-11-01470-f002:**
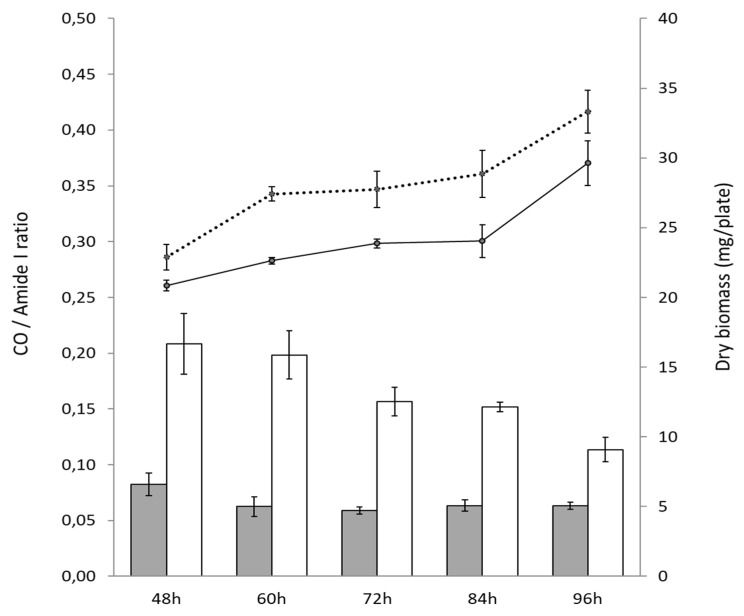
FTIR analysis: CO/Amide I ratio of *S. coelicolor* M145 (grey histograms) and its TD variant (white histograms) at different points throughout the cultivation period carried out at 28 °C on solid R2YE medium limited in Pi (1 mM Pi) and assessment of dry biomass of *S. coelicolor* M145 (black continuous line) and its TD variant (black dotted line) at these time points.

**Figure 3 microorganisms-11-01470-f003:**
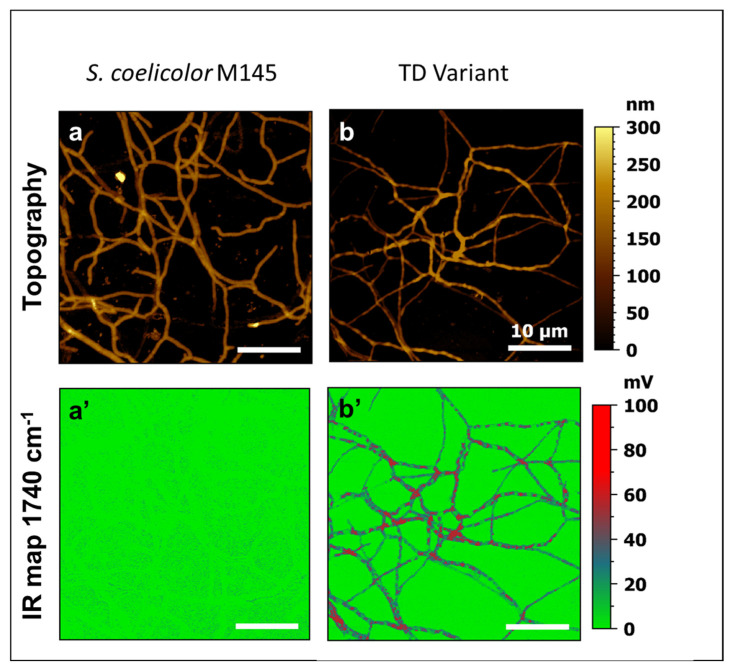
Comparative AFM-IR analysis of *S. coelicolor* M145 (left) and its TD variant (right). Topography images (**a**,**b**) and AFM-IR maps at 1740 cm^−1^ (**a’**,**b’**). IR maps at 1740 cm^−1^ revealed the presence of red patches corresponding to TAG vesicles in the cytoplasm of the TD variant that were absent from the cytoplasm of the original strain of *S. coelicolor* M145.

**Figure 4 microorganisms-11-01470-f004:**
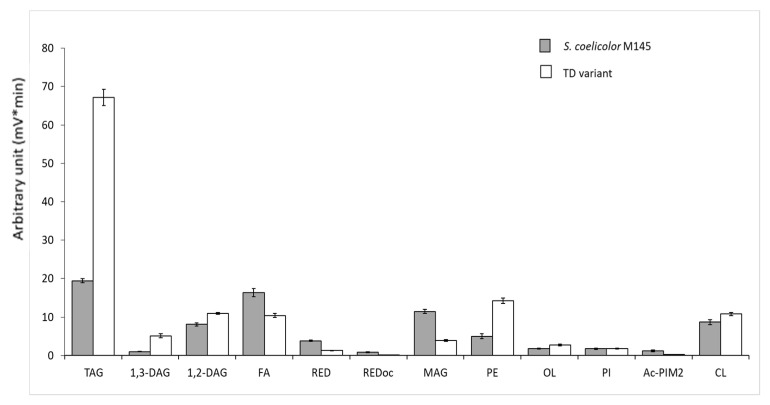
LC/Corona-CAD analysis of the total lipid content of *S. coelicolor M145* (grey histograms) and its TD variant (white histograms) grown for 72 h at 28 °C on solid R2YE medium limited in Pi. TAG/triacylglycerol; DAG/diacylglycerol (1.2 or 1.3); FA/fatty acids; RED/undecylprodigiosin, MAG/monoacylglycerol; PE/phosphatidylethanolamine; OL/ornithine lipids; PI/phosphatidylinositol; Ac-PIM2/acetylated phosphatidylinositol mannoside 2; CL/cardiolipid. Mean values of three replicates are shown as histograms with error bars representing standard error (*p* > 0.05; Tukey-adjusted comparisons).

**Figure 5 microorganisms-11-01470-f005:**
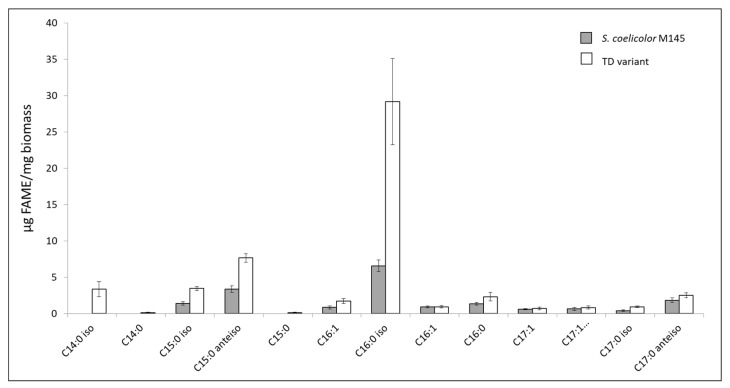
Quantification of fatty acid methyl esters (FAMEs) in *S. coelicolor* M145 (grey histograms) and its TD variant (white histograms) determined by GC-MS. Mean values of three replicates are shown as histograms with error bars representing standard error (*p* > 0.05; Tukey-adjusted comparisons). The letter C followed by the numbers 14, 15, 16 or 17 represents the number of carbons present in the fatty acid. The numbers, 0 or 1, following the sign “:” indicate the number of unsaturations (double bonds) present in the fatty acid. Iso- fatty acids have a methyl group branched on the penultimate carbon of the carbon chain while anteiso fatty acids have a methyl group branched on the ante-penultimate carbon atom of the carbon chain.

## Data Availability

The sequencing reads of the genome of the TD variant (raw data) were submitted to SRA under the reference SUB12982887, on 23 March 2023.

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
