# Peer review of "Genome Analysis of a Variant of *Streptomyces coelicolor* M145 with High Lipid Content and Poor Ability to Synthetize Antibiotics"

_microorganisms, 2023, doi:10.3390/microorganisms11061470_

Round 1
Reviewer 1 Report
Dulermo et al. analyzed the mutant which was found while preparing the targeted deletion mutant of sco0982. The mutant, named TD, showed a reduced production of ACT, and high content of lipids (fatty acids) in the cell as demonstrated by FTIR, ATM-IR, LC/Corona-CAD, and GC/MS. The genome of mutant TD was re-sequenced to identify the mutation(s). Although the mechanism of lipid accumulation in mutant TD is still unclear due to numerous deletions of the gene locus, the results provide additional support for the phenomenon of a negative correlation between total lipid content and antibiotic production in Streptomyces species. As the experimental contents seems clear, I support the publication of this manuscript in Microorganisms after several revisions.
Major issues:
Figure 1. > contains inconsistent data.
Figure 1A only shows the quantification results from R2TE with 1 mM Pi. Figure 1B only shows photo images of plate grown on R2YE with 5 mM Pi. Please present photo images of plate grown on R2YE with 1 mM Pi, or add explanation in the figure legend.
Table S2:
There is list in Sheet1. But I believe there are no explanation or description on them.
Minor issues:
L123-125: Figure indications are not appropriate.
L113: et > and
L116: ErmE > ermE
L213: that > than
L348: remove “encoding”
L384: sub-units > remove hyphen.
L387: 1rst > first.
L391: > mentioned
Author Response
We are grateful to the reviewer for his/her comments
Comments of the reviewer were all taken into account and all changes are highlighted in yellow in the revised version of the manuscript.
Major issues:
Figure 1. > contains inconsistent data.
Figure 1A only shows the quantification results from R2TE with 1 mM Pi. Figure 1B only shows photo images of plate grown on R2YE with 5 mM Pi. Please present photo images of plate grown on R2YE with 1 mM Pi, or add explanation in the figure legend.
The figure legend was modified to fulfill reviewer requirement.
Table S2:
There is list in Sheet1. But I believe there are no explanation or description on them.
Yes the legend of Table S2 was present in the submittted manuscript p 14 and is now highlighted in yellow.
Minor issues:
L123-125: Figure indications are not appropriate.
This was modified.
Comments on the Quality of English Language
L113: et > and Correction done
L116: ErmE > ermE Correction done
L213: that > than Correction done
L348: remove “encoding” Encoding was removed
L384: sub-units > remove hyphen. Correction done
L387: 1rst > first. Correction done
L391: > mentioned Correction done
Reviewer 2 Report
In this study, the authors found an unexpected variant (TD) of Streptomyces coelicolor M145 from the sco0982 deletion process has a 3 fold higher triacylglycerol and phosphatidylethanolamine content than the original strain. They further characterized this strain and investigated the possible reason through comparative genomics.
Major comments:
1, The author guessed that oxidative stress from deletion of sco0982 resulted in the genome instability of S. coelicolor M145. How many strains did exhibit the genomic instability during deleting sco0982?
2, Please indicate which gene or gene deletion is the key point for phenotypic changes in this strain?
3, Please check the concentration of cellular ROS. It could tell whether oxidative stress result in the phenotypic changes in TD.
Minor comments:
1, Lines 212-213, should be “the TD variant reproductively yielded higher biomass than the original strain”.
Author Response
Answers to reviewer 2 comments are in the joined file.

Reviewer 3 Report
In this paper, the authors describe the analysis of the genome of Streptomyces coelicolor M145 model strain TD variant, which contains an increased amount of lipids with reduced production of secondary metabolites. During the production of the mutant in the gene that codes for enzyme isocitrate lyase, the authors also obtained phenotypically specific colonies of mutants with reduced production of secondary metabolites, actinorhodin, with increased production of total lipids. The authors characterized the phenotypes of the obtained mutant and further sequenced the genome of this mutant. The results of the genome analysis showed the success of the inactivation of the desired gene, but also a large number of deletions in the genome of this bacterium. The authors provide a detailed analysis of the genes that were "deleted" from the genome of a new mutant that shows great genetic instability. Conclusions about the number of missing genes should be taken with a certain reserve. The paper lacks more information about the results of the sequenced genome, the number of contigs, gaps and point mutations. The authors provide the genome acc number in the data bank, however, the sequence is not available for detailed analysis.
The deletions of individual genes that the authors mention can be in the part of the genome that is not sequenced (gaps).
The authors comment on certain genes that are absent in the genome of this mutant, unfortunately they do not comment on the biosynthesis of mevalonate as a common precursor in the biosynthesis of secondary metabolites.
The authors do not comment on metabolism via methyl citrate and production of succinate from propionate.
The mutation of propionyl-CoA carboxylase could stop the metabolic pathway towards the formation of succinate and probably increased the concentration of acyl-CoA (propionyl-CoA) which results in increased lipid biosynthesis. This possibility is in accordance with the increased amount of C15 and C17 fatty acids. If the original strain has no methylmalonyl-CoA racemase and methylmalonyl-CoA mutase genes authors should state this and include it in discussion.
Deletion of methylase gene is very interesting as candidate gene to explain instability of TD variant genome.
Somme comments :
Line 190. Table 1 is Supplementary Table 1.
Line 118; Specific E coli strain replace with name of strain.
In general, all abbreviations must be explained the first time they appear.
Moderate editing .
Author Response
We first wish to thank this reviewer for her/his valuable comments that we took into account in our revised manuscript (version 2).
In this paper, the authors describe the analysis of the genome of Streptomyces coelicolor M145 model strain TD variant, which contains an increased amount of lipids with reduced production of secondary metabolites. During the production of the mutant in the gene that codes for enzyme isocitrate lyase, the authors also obtained phenotypically specific colonies of mutants with reduced production of secondary metabolites, actinorhodin, with increased production of total lipids. The authors characterized the phenotypes of the obtained mutant and further sequenced the genome of this mutant. The results of the genome analysis showed the success of the inactivation of the desired gene, but also a large number of deletions in the genome of this bacterium.
In the TD mutant the deletion of the gene encoding the isocitrate lyase is included in a large deletion.
The authors provide a detailed analysis of the genes that were "deleted" from the genome of a new mutant that shows great genetic instability.
We think that high genetic instability is indeed at the origin of the generation of the TD mutant but the latter is most probably far more stable than the original strain since it has lost 2/3 of its numerous insertion sequences.
Conclusions about the number of missing genes should be taken with a certain reserve. The paper lacks more information about the results of the sequenced genome, the number of contigs, gaps and point mutations. The authors provide the genome acc number in the data bank, however, the sequence is not available for detailed analysis. The deletions of individual genes that the authors mention can be in the part of the genome that is not sequenced (gaps).
Sequencing data are usually released on the date of publication, but we have specifically asked NCBI/SRA to release them immediately. This is a request that is currently being processed.
We agree with the referee about the level of caution that should be taken in asserting that a sequence is deletion by the approach we have used. As stated in the materials and methods and in the text, we proceeded by aligning the sequencing reads of the mutant to the NCBI reference of S. coelicolor. In order to avoid the pitfall of unsequenced regions, we then confronted the suspected deletion regions (not covered by the alignment) with the contigs provided by the provider (contigs not provided on SRA), and confirmed that these suspected deletions did not correspond to sequencing gaps.
Moreover, the gaps linked to unsequenced areas are classically regions with a high proportion of GC bases, which correspond to regions of very small size (a few tens of nt) that we can be otherwise identified on the reference. In other words, we can distinguish with a good level of confidence a deletion of one or more genes (covering several kilobases) and a non-sequenced area (a few tens at most).
Moreover, the high coverage rate obtained during the collection of reads allows us to be confident that when a region of several kilobases is not covered, it is indeed absent from the genome of the TD mutant.
The authors comment on certain genes that are absent in the genome of this mutant, unfortunately they do not comment on the biosynthesis of mevalonate as a common precursor in the biosynthesis of secondary metabolites.
The mevalonate pathway has not been clearly identified in S. coelicolor. Some genes encoding putative geranylgeranyl pyrophosphate synthase (sco0185), HMG-CoA lyase (sco2778), isopentenyl mono phosphate kinase (sco3148) and IPP isomerase (sco6750) could be found in this genome but these genes are scattered in the genome and none of them is deleted in the TD variant.
The authors do not comment on metabolism via methyl citrate and production of succinate from propionate. The mutation of propionyl-CoA carboxylase could stop the metabolic pathway towards the formation of succinate and probably increased the concentration of acyl-CoA (propionyl-CoA) which results in increased lipid biosynthesis. This possibility is in accordance with the increased amount of C15 and C17 fatty acids. If the original strain has no methylmalonyl-CoA racemase and methylmalonyl-CoA mutase genes authors should state this and include it in discussion.
The S. coelicolor genome includes 4 genes annotated as encoding putative (methyl) citrate synthases (sco5832, sco2736, sco4388, sco5831) that might belong to a putative methyl citrate pathway but none of these are deleted in the TD variant. So we think that this putative pathway, that is not clearly identified, in Streptomyces is still functional in the TD mutant.
The S. coelicolor genome includes several genes clearly annotated as propionate carboxylase (sco2776_77, sco4380_81, sco4626, sco5335 that might catalyze the conversion of propionyl-CoA into D-methylmalonyl-CoA. None of these genes is deleted. In the TD mutant.
Some putative methylmalonyl-CoA racemase (conversion D to L form) encoding genes were predicted in S. coelicolor (sco0362, sco2032, sco6730, sco6785…) but none of them was deleted. Two genes annotated as encoding methylmalonyl-CoA mutase, sco4869 and sco6832, that might catalyze the conversion of L-methylmalonyl-CoA into succinyl-CoA are also present in the genome of S. coelicolor but sco6832 is deleted in the TD variant.
So the reviewer might be right if one considers that the deletion of sco6832 in the TD variant has a great negative impact on the generation of succinylCoA so a few lines on this matter were added at the end of page 12 in the revised version 2 of the manuscript.
Deletion of methylase gene is very interesting as candidate gene to explain instability of TD variant genome.
The following DNA methylase encoding genes sco5331 embedded in a large deletion sco5327-sco5350, sco6844 embedded in a large deletion sco6806-sco6953 as well as sco6885 were deleted in the TD variant. Numerous reports in the litterature mentionned that DNA hypomethylation is correlated with genomic instability. So we agree that these deletions could play a role in the genetic instability that led to the generation of the TD variant so we added a few lines on this matter p14 of the revised manuscript.
Some comments :
Line 190. Table 1 is Supplementary Table 1.
Correction done.
Line 118; Specific E coli strain replace with name of strain.
Correction done highlighted in green p 4
In general, all abbreviations must be explained the first time they appear.
Correction done highlighted in green p 2.
Round 2
Reviewer 2 Report
The authors have answered all the question, I have no comment.